# Prognostic Values of Systemic Inflammatory Immunological Markers in Glioblastoma: A Systematic Review and Meta-Analysis

**DOI:** 10.3390/cancers15133339

**Published:** 2023-06-25

**Authors:** Pawel Jarmuzek, Klaudia Kozlowska, Piotr Defort, Marcin Kot, Agnieszka Zembron-Lacny

**Affiliations:** 1Department of Nervous System Diseases, Collegium Medicum University of Zielona Gora, Neurosurgery Center University Hospital in Zielona Gora, 65-417 Zielona Gora, Poland; p.jarmuzek@cm.uz.zgora.pl (P.J.); m.kot@cm.uz.zgora.pl (M.K.); 2Department of Biomedical Engineering, Faculty of Fundamental Problems of Technology, Wroclaw University of Science and Technology, 50-370 Wroclaw, Poland; klaudia.kozlowska@pwr.edu.pl; 3Department of Applied and Clinical Physiology, Collegium Medicum University of Zielona Gora, 65-417 Zielona Gora, Poland; a.zembron-lacny@cm.uz.zgora.pl

**Keywords:** brain tumor, cell-free DNA, inflammation, neutrophils, platelets, prognosticators

## Abstract

**Simple Summary:**

The authors report the most up-to-date review and a thorough meta-analysis of inflammatory, immunological markers such as neutrophil-to-lymphocyte ratio (NLR), platelet-to-lymphocyte ratio (PLR), systemic immune inflammation index (SII) and systemic inflammation response index (SIRI) as prognostic factors in patients with glioblastoma. A number of studies showed the important prognostic value of inflammatory immune markers. Similarly, some studies reported on the potential benefits of the measurements of small cell-free DNA fragments (cfDNA) released into the bloodstream as the biomarker of early diagnosis or/and prognosis. Twenty-one studies met our meta-analysis criteria assessing the prognostic significance of NLR, PLR, SII, SIRI, and cfDNA. According to our findings, NLR, PLR, and cfDNA fare significantly better than SII and SIRI in the evaluation of prognosis in glioblastoma patients. NLR and PLR calculated from routine blood tests, potentially in combination with measurements of cfDNA, can help assess disease progression and optimize treatment and follow-up.

**Abstract:**

Background. Neutrophils are an important part of the tumor microenvironment, which stimulates inflammatory processes through phagocytosis, degranulation, release of small DNA fragments (cell-free DNA), and presentation of antigens. Since neutrophils accumulate in peripheral blood in patients with advanced-stage cancer, a high neutrophil-to-lymphocyte ratio can be a biomarker of a poor prognosis in patients with glioblastoma. The present study aimed to explore the prognostic value of the preoperative levels of neutrophil-to-lymphocyte ratio (NLR), platelet-to-lymphocyte ratio (PLR), systemic immune inflammation index (SII), systemic inflammation response index (SIRI), and cell-free DNA (cfDNA) to better predict prognostic implications in the survival rate of glioblastoma patients. Methods. The meta-analysis was carried out according to the recommendations and standards established by the Preferred Reporting Items for Systematic Reviews and Meta-Analyses. Databases of PubMed, EBSCO, and Medline were systematically searched to select all the relevant studies published up to December 2022. Results. Poorer prognoses were recorded in patients with a high NLR or PLR when compared with the patients with a low NLR or PLR (HR 1.51, 95% CI 1.24–1.83, *p* < 0.0001 and HR 1.34, 95% CI 1.10–1.63, *p* < 0.01, respectively). Similarly, a worse prognosis was reported for patients with a higher cfDNA (HR 2.35, 95% CI 1.27–4.36, *p* < 0.01). The SII and SIRI values were not related to glioblastoma survival (*p* = 0.0533 and *p* = 0.482, respectively). Conclusions. Thus, NLR, PLR, and cfDNA, unlike SII and SIRI, appeared to be useful and convenient peripheral inflammatory markers to assess the prognosis in glioblastoma.

## 1. Introduction

In the last few years, it has become clear that tumor-associated inflammation progresses from acute to chronic inflammation. Acute inflammation is a protective response caused by injury or infection, while chronic inflammation supports immunosuppression, and the inflammatory process can increase tumor cell proliferation and survival [1,2]. Abnormal activation of inflammatory responses is an essential feature of glioblastoma (GBM), which allows tumor cells to evade a response of the immune system, leading to immune tolerance of GBM to therapy [3]. Neutrophils and lymphocytes are classic inflammatory cells, and their elevated counts in peripheral blood are associated with increased inflammation. Similar to the majority of cancer patients, most patients with gliomas have robust neutrophilia and lymphopenia caused by an overproduction of granulocyte colony-stimulating factor (G-CSF) by tumor cells [4]. The function of neutrophils remains controversial as they were shown to have both tumor-promoting and limiting properties. The circulating and tumor-associated neutrophils (TANs) are not a homogeneous population as previously considered [5]. TANs are involved in the tumor microenvironment via cytokines and chemokines and can be distinguished according to their activation, cytokine status, and the effects that cytokines produce on the pro-tumor and anti-tumor functions of neutrophils. Anti-tumor activity is revealed by direct or indirect cytotoxicity, whereas pro-tumor neutrophils stimulate immunosuppression, tumor growth, angiogenesis, and metastasis through DNA instability or via cytokine and chemokines activity [6]. In patients with GBM, a high number of neutrophils and a high neutrophil-to-lymphocyte ratio (NLR) correlate with a poor prognosis; therefore, neutrophil count and NLR are considered as onco-inflammatory markers [7].

Glioblastoma is the most aggressive primary malignant brain tumor in adults, with a median survival time of 15–23 months and a five-year survival rate lower than 6% after initial diagnosis, even if GBM patients have received standard treatments, including surgery, radiotherapy, and chemotherapy [8]. A total of 90% of GBMs primarily occur in older patients, while in younger patients’ tumors tend to progress from lower-grade glioma. The fifth edition of the World Health Organization Classification of Tumors of the Central Nervous System (WHO CNS5), published in 2021, established new tumor types and subtypes based on novel diagnostic technologies such as genome-wide profiling of DNA methylome [9]. WHO CNS5 has incorporated numerous molecular changes with clinicopathological utility that are important for the most accurate classification of the central nervous system (CNS) neoplasms based on the key genes and proteins that are analyzed for diagnostic changes important for the integrated classification of CNS tumors. However, WHO CNS5 has not recommended the molecular evaluation of the individual diagnostic lesions unless this method is clearly required for the diagnosis of a distinct tumor type or subtype [9]. Therefore, it has become a matter of urgency to find additional and easily testable markers to predict the outcomes in glioma patients.

Assessment of inflammatory processes by conventional blood tests is often beneficial in the diagnosis of early stages of diseases as well as in the clinical prognosis of brain tumors [7,10,11,12,13,14,15,16,17,18,19]. According to Massara et al. [5], NLR higher than 4 was associated with poor prognosis when measured before standard treatments. NLR lower than 4 was associated with better prognosis but only in GBM expressing the wild-type gene isocitrate dehydrogenase 1 (IDH1), one of the genes which is more frequently mutated in malignant gliomas [20]. Additionally, a standard blood count is easy, inexpensive, and delivers information on a variety of cell types together with morphological parameters, i.e., leucocytes, lymphocytes, neutrophils, monocytes, and platelet count. Some studies reported that a combination of hematological components, such as the neutrophil-to-lymphocyte ratio, the platelet-to-lymphocyte ratio (PLR), the lymphocyte-to-monocyte ratio (LMR), the systemic immune inflammation index (SII) and the systemic inflammation response index (SIRI) were effective prognostic markers in patients with a variety of cancers [21,22,23,24,25,26,27]. A comparison made between these hematological markers and traditional molecular prognostic markers demonstrated that isocitrate dehydrogenase-1 mutation in gliomas was associated with chronic low-grade inflammation, which could be associated with a better prognosis in this subgroup of patients [28]. Recently, four meta-analyses demonstrated that NLR could be considered a prognostic factor in GMB, and some modification of chemotherapy should be recommended in high-risk patients [29,30,31,32]. In our retrospective study, we turned our attention to systemic inflammatory-immune markers, NLR, SII, and SIRI, which are based on neutrophil counts and which all exceeded the reference values proposed by Luo et al. [33] and Qui et al. [34] (0.87–4.15 103/µL for NLR, 142–808 103/µL for SII and 0.41–1.42 103/µL for SIRI). The Cox model analysis showed that NLR ≥ 4.56 × 103/µL, SII ≥ 2003 × 103/µL, and SIRI ≥ 3.03 × 103/µL significantly increased the risk of death in GBM patients [7].

Some studies have highlighted the potential benefits of the measurement of small cell-free DNA fragments (cfDNA), which are released from the tumor and healthy cells into the bloodstream as a result of secretion, apoptosis, necrosis, or NETosis [35,36,37]. A total of 85% of plasma cfDNA fragments in cancer patients are 166 base pair (bp), 10% are 332 bp, and 5% are 498 bp in length. Larger cell-free DNA fragments, i.e., ~10,000 bp in length, are the products of necrosis. The elevated levels of cfDNA have been documented in malignant tumors among adults, including glioblastoma patients, relative to patients with non-neoplastic diseases [35]. However, the levels of cfDNA in brain tumors are reduced by 60% in medulloblastoma, and by 90% in low-grade glioma, as compared to systemic malignancies [35,36,37]. Therefore, the measurement of cfDNA in glioblastoma patients for clinical applications remains a multifaceted problem. Therefore, circulating cell-free DNA and tumor-derived DNA fraction are currently analyzed in the context of a liquid biopsy and blood samples as they appear to be promising potential biomarkers for the early diagnosis or prognosis in glioblastoma [38,39,40]. So far, only one available meta-analysis by MacMahon et al. [36] showed that cell-free DNA appears to be a significantly sensitive and specific biomarker in adults with low- and high-grade gliomas; however, further studies should be conducted with glioblastoma as a target.

Inflammation is a key to understanding GBM development, and anti-inflammatory treatment may be one of the ways to reduce the risk. Therefore, it has become extremely urgent to evaluate the inflammatory profile to predict the outcomes in patients with GBM. The aim of the study was to carry out the meta-analysis with a view to systematically evaluating inflammatory immune markers and presenting a deeper understanding of the prognostic value of NLR, PLR, SII, SIRI, and cell-free DNA in adults with glioblastoma multiforme.

## 2. Materials and Methods

### 2.1. Search Strategy

The meta-analysis was carried out according to the recommendations and standards established by the Preferred Reporting Items for Systematic Reviews and Meta-Analyses (PRISMA) [41]. We carried out a comprehensive Internet literature search of the following English databases: PubMed/Medline, Embase, Web of Science, and Cochrane Library. The queries were last updated on 20 December 2022. The search terms contained a combination of the following phrases: “Glioblastoma” and “NLR/PLR/SII/SIRI/cell-free DNA”. Table 1 presents the PubMed/Medline search strategy using MeSH terms.

### 2.2. Selection Criteria

The first step of selection was the title of the publication and the selection of its abstract. We excluded: (1) cohort studies, case reports, case series, letters, conference abstracts, reviews, and books, (2) non-English records, and (3) duplicated publications. Reviewers independently evaluated the eligibility of the article, and any disagreement was resolved through discussion.

### 2.3. Eligibility Criteria

The following inclusion criteria were applied for our meta-analysis: (1) patients confirmed with glioblastoma multiforme, (2) evaluation of peripheral blood NLR/PLR/SII/SIRI/cell-free DNA, (3) provided the prognostic significance of peripheral blood NLR/PLR/SII/SIRI/cell-free DNA, (4) provided cut-off values of NLR/PLR/SII/SIRI/cell-free DNA and (5) hazard ratio (HR), 95% confidence intervals (CI), and overall survival (OS).

### 2.4. Data Extraction and Quality Assessment

Each article was described by providing the following details: the first author and year of publication, study duration, number of patients with respect to sex, cut-off values for NLR/PLR/SII/SIRI/cell-free DNA, univariate and multivariate outcome, and type of glioblastoma. When both univariate HR and multivariate HR were reported, only multivariate HR was used. We used the Newcastle–Ottawa Scale (NOS) to assess the quality of the studies [42]. NOS evaluates the following points: patient selection, comparability of study groups, and evaluation of results. We defined high-quality studies with scores of at least seven out of nine stars.

### 2.5. Statistical Analysis

Statistical analyses were using R 4.2.1 software (https://www.r-project.org/, accessed on 20 December 2022) and a “meta” package using a random effects model [43]. Chi-squared and Higgin’s I^2^ tests were used to measure heterogeneity between studies. Cut-off values of 25%, 50%, and 75% were applied to label heterogeneity as low, moderate, or high [44]. For I^2^ > 50%, a subgroup analysis was attempted in relation to the type of glioblastoma (glioblastoma IV grade, glioblastoma multiforme, and IDH mutation) to find the source of heterogeneity. Funnel plots were used to assess publication bias. In case asymmetry was present in the funnel plots, Egger’s and Begg’s tests were applied to quantitatively assess whether there was publication bias. We performed the sensitivity analysis by excluding a single study from the analysis to examine the stability of the results. The significance threshold for all statistical tests was set at *p* < 0.05.

## 3. Results

### 3.1. Study Search and Characteristics

The detailed search selection of studies for the meta-analysis is presented in Figure 1. A total of 487 studies were retrieved from the initial search. After removing duplicates, 437 studies were screened. After screening the titles and abstracts, we excluded 387 records. A total of twenty-one full-text manuscripts published between 2013 and 2022 were examined.

We collected the data from 2743, 1171, 1405, 866, and 104 patients in whom the prognostic significance of NLR, PLR, SII, SIRI, and cell-free DNA, respectively, were assessed. The majority of the reviewed studies performed a multivariate Cox regression analysis and reported adjusted HR. The main characteristics of the selected studies are shown in Table 2. The NOS scoring details are presented in Table 3. NOS scores ranged from 7 to 8. The average number of NOS scores was 7.6.

### 3.2. Analysis of NLR

The outcomes of fifteen studies comprising 2743 patients showed that patients with a higher NLR had a worse prognosis (HR: 1.51, 95% CI (1.24; 1.83), *p* < 0.0001, I^2^ = 86.04%, 95% CI (79.10%; 90.67%)) (Figure 2). The quality of the evidence was moderate. The asymmetry of the funnel plot was observed (Figure 3) (*p* = 0.007). Sensitivity analysis showed that the result was stable. Two subgroups, with glioblastoma grade IV and IDH mutation patients, could be a potential source of high heterogeneity (I^2^ equal to 76.56% and 88.00%, respectively). Due to a high variation between groups, the effect observed for the IDH mutation subgroup was found insignificant (Figure 4).

### 3.3. Analysis of PLR

Nine studies with a total of 1171 included patients reported data on overall survival with regard to PLR. Unfavorable OS was observed in patients with a high PLR (HR: 1.34, 95% CI (1.10; 1.63), *p* = 0.01, I^2^ = 34.88%, CI (0.00%; 72.76%)) (Figure 5), with a moderate quality of supporting evidence. The asymmetry of the funnel plot was not observed (Figure 6). Sensitivity analysis showed that the result was stable. One subgroup, i.e., GBM patients, could be a potential source of heterogeneity (I^2^ equal to 71.04%) (Figure 7).

### 3.4. Analysis of SII

Six studies comprising 1405 patients in total reported the relationship between overall survival and SII. Worse prognoses were not observed in patients with a higher SII (HR: 1.34, 95% CI (1.00; 1.81), *p* = 0.05, I^2^ = 81.21%, 95% CI (59.70%; 91.24%)) (Figure 8). The quality of the evidence was low. The asymmetry of the funnel plot was observed (Figure 9) (*p* = 0.8). Sensitivity analysis showed that the result was stable. Analysis of two subgroups showed that glioblastoma grade IV contributed most to the final heterogeneity (I^2^ equal to 84.06%) (Figure 10).

### 3.5. Analysis of SIRI

The review of the outcomes reported in five studies comprising 866 patients showed that patients with a higher SIRI did not have a worse prognosis (HR: 1.16, 95% CI (0.77; 1.73), *p*~0.05, I^2^ = 86.29%, 95% CI (70.09%; 93.71%)) (Figure 11). The quality of the evidence was low. The asymmetry of the funnel plot was observed (Figure 12). Sensitivity analysis showed that the result was stable. We conducted a second analysis only for glioblastoma grade IV [7] and obtained a non-significant overall effect (HR: 1.03, 95% CI (0.62; 1.71), *p* = 0.09).

### 3.6. Analysis of cfDNA

The outcomes of two studies comprising 104 patients showed a worse prognosis for patients with a higher cfDNA value (HR: 2.35, 95% CI (1.27; 4.36), *p* = 0.007, I^2^ = 0.00%) (Figure 13). The source of heterogeneity was unknown. Since only two relevant studies were reviewed here, sensitivity and publication bias analysis was not performed.

## 4. Discussion

Systemic chronic inflammation is one of the features of tumorigenesis. Inflammation predisposes the development of cancer and promotes all stages of tumorigenesis, from cell transformation to metastasis [7]. Extracranial glioblastoma metastases are extremely rare, affecting only 0.4–0.5% of all patients with GBM. Nevertheless, inflammatory evaluation seems to be crucial in our understanding of the highly aggressive nature of GBM [62]. In this study, we performed a meta-analysis to evaluate the prognostic role of systemic inflammatory markers calculated in peripheral blood (NLR, PLR, SII, SIRI) in 6289 GBM patients reported in 21 available papers. Our study confirmed the conclusion drawn previously that some of the indicators could be valuable prognostic markers in patients with GBM [29,63,64]. Increased levels of NLR and PLR were found to be independent predictors of worse survival in GBM. The pooled HR was considered significant if the 95% CI did not overlap 1, and the *p*-value was <0.05. A poorer prognosis was observed in the patients with a high NLR or PLR in comparison with the patients with a low NLR or PLR values (HR 1.51, 95% CI 1.24–1.83, *p* < 0.0001 and HR 1.34, 95% CI 1.10–1.63, *p* < 0.01, respectively). We also evaluated the SII and SIRI, but neither of these variables correlated with the overall survival rate (*p* = 0.0533 and *p* = 0.482, respectively). Consequently, it is NLR and PLR that may serve as potential prognosticators for survival outcomes.

The NLR comprises the peripheral neutrophil and lymphocyte counts, thereby indicating the balance between the inflammation and immune responses in GBM [7]. Most glioma patients experience severe neutrophilia and lymphopenia due to the high production of granulocyte colony-stimulating factors by tumor cells [5]. The actual function of neutrophils remains controversial, as they were previously shown to possess both tumor-promoting and tumor-limiting properties [65]. Neutrophils participate in the formation of an inflammatory environment, being the main source of interleukin 6, interleukin 8, transforming growth factor β, hepatocyte growth factor, and matrix metalloproteinases [66,67,68]. Neutrophils are also responsible for releasing the factors of tumor-related angiogenesis, including vascular endothelial growth factor, fibroblast growth factor-2, and angiopoietin-1 [69]. A growing body of evidence shows that neutrophils play an important role in different stages of tumor development and that the neutrophils count is an early predictor of tumor progression in patients with glioblastoma [70]. An elevated number of neutrophils leads to a decrease in lymphocyte count, inducing T-cell apoptosis by the interaction of neutrophil PD-L1 with its lymphocyte receptor PD-1. Moreover, neutrophils can directly reduce lymphocyte proliferation through the production of arginase-1 and hydrogen peroxide [71]. Lymphocytes, in turn, conduct immune surveillance and are found to be protective prognostic factors for cancer patients [72]. Higher levels of CD4+ T lymphocytes can reduce the risk of recurrence, while lower levels of CD4+, CD8+, CD3+, and CD56+ T cells in advanced tumors reduce the lymphocyte-mediated anti-tumor cellular immune response, ultimately leading to worse prognosis [51]. The direct neutrophil modulation of T-cell effector functions, in combination with the central role of T cells in immune surveillance and tumor cell destruction likely to be responsible for the mechanism by which NLR appears to be a valuable prognostic marker [71]. In our retrospective study, we observed a poorer prognosis in the patients with NLR ≥ 4.56 × 103/μL [7], and the meta-analysis of fifteen studies, which included 2743 patients, confirmed this result (HR 1.51, 95% CI 1.24–1.83, *p* < 0.0001).

The PLR has already been proven to be important in the development and progression of a number of tumors, such as gastric cancer, gastrointestinal stromal tumors, lung cancer, and renal cell carcinoma, whereas its prognostic role in glioblastoma remains undetermined [73,74,75,76]. A high preoperative PLR level was reported as a predictor of poor prognosis for GBM patients [77]. Our meta-analysis of nine studies with 1171 patients demonstrated that patients with high PLR had an unfavorable overall survival rate (HR 1.34, 95% CI 1.10–1.63, *p* < 0.01). Elevated platelets may enhance tumor growth and angiogenesis by secreting VEGF [78]. However, some studies demonstrated that pretreatment PLR was not associated with improved overall survival of patients with GBM [79]. According to Wang et al. [70], the discrepancy may result from the differences in sample size and cut-off values. Han et al. [46] reported that the PLR carried far less prognostic significance than NLR in a study of GBM. A high PLR is observed prior to surgery in a variety of intracranial neoplasms, but PLR does not differentiate glioma from meningioma or acoustic neuroma [23,78]. Despite the fact that relatively high levels of PLR were found in patients with glioblastoma, the underlying explanation still needs further investigation [29,70].

The SII and SIRI, based on neutrophils, lymphocyte monocytes, and platelet counts, were indicated as prognosticators of the risk of death in GBM [7]. Topkan et al. [79] demonstrated a significant association between a low SIRI and longer progression-free and overall survival durations in the multivariate Cox analysis. The authors suggested that SIRI may be a novel and independent predictor of survival in newly diagnosed GBM patients who underwent postoperative treatment according to the Stupp protocol. The present meta-analysis was focused on the preoperative period and did not confirm the prognostic importance of SII and SIRI (HR 1.34, 95% CI 1.00–1.81, *p* > 0.05 and HR 1.16, 95%CI 0.77–1.73, *p* > 0.05, respectively). Therefore, we adhere to the prognostic significance of pre-treatment neutrophil-to-lymphocyte ratio and platelet-to-lymphocyte ratio in glioblastoma progression and/or survival probability.

We also evaluated cell-free DNA, but only two studies that met the selection criteria were reviewed; therefore, the results should be applied with caution. Cell-free DNA is the pool of circulating genetic material derived from various cells that release DNA fragments in the process of cell death. Circulating tumor DNA, the subset of cfDNA shed from tumor cells, has recently become a promising marker in patients with advanced cancer. The meta-analysis performed by MacMahon et al. [36] revealed better biomarker performance for cerebrospinal fluid ctDNA (AUC = 0.947) compared to plasma ctDNA (AUC = 0.741). Our meta-analysis of two studies comprising 104 patients demonstrated a worse prognosis in patients with a higher cfDNA (HR 2.35, 95%CI 1.27–4.36, *p* < 0.01). One of these studies reported a significant difference in plasma cfDNA content in patients with glioblastoma vs. healthy control and a worse prognosis in patients with glioblastoma with higher cfDNA concentrations [52]. The fragments of DNA released by apoptotic and necrotic cells are termed circulating tumor DNA (ctDNA). The total amount of ctDNA represents only 0.1–5% of total cell-free DNA and is tightly correlated with tumor type and its grading [80]. Circulating tumor DNA exhibits genetic alterations such as single nucleotide variants, gene copy number variations, or chromosomal rearrangements. Nevertheless, because of the low concentration of ctDNA, the detection technology has to be highly sensitive and specific to distinguish ctDNA from normal leucocyte DNA. The mean half-life of ctDNA is short and ranges between 1.5 and 2 h, which altogether makes the detection of ctDNA extremely challenging [81]. Therefore, total cell-free DNA seems to be a simpler and more useful biomarker for adults with glioblastoma. Accurate circulating biomarkers have potential clinical applications in population screening, tumor subclassification, tumor status monitoring, and individualized treatments based on tumor genotyping [80,82,83].

## 5. Conclusions

Our meta-analysis indicates the high diagnostic usefulness of peripheral immune-inflammatory markers NLR and PLR over SII and SIRI in the prognosis of patients with GBM. Pre-operative NLR and PLR assessment can help to evaluate disease progression, optimize treatment, or introduce anti-inflammatory agents and follow-up patients with GBM. However, further prospective studies are needed to verify the reliability of the meta-analysis performed, especially with regard to the circulating cell-free DNA.

## 6. Limitations

Several limitations to this meta-analysis should be acknowledged. Firstly, the analyzed studies included in the meta-analysis involved either univariate or multivariate HR results., Secondly, the studies did not take into account the size and different stages of tumor differentiation. Finally, the quality of evidence ranged from low to moderate.

## Figures and Tables

**Figure 1 cancers-15-03339-f001:**
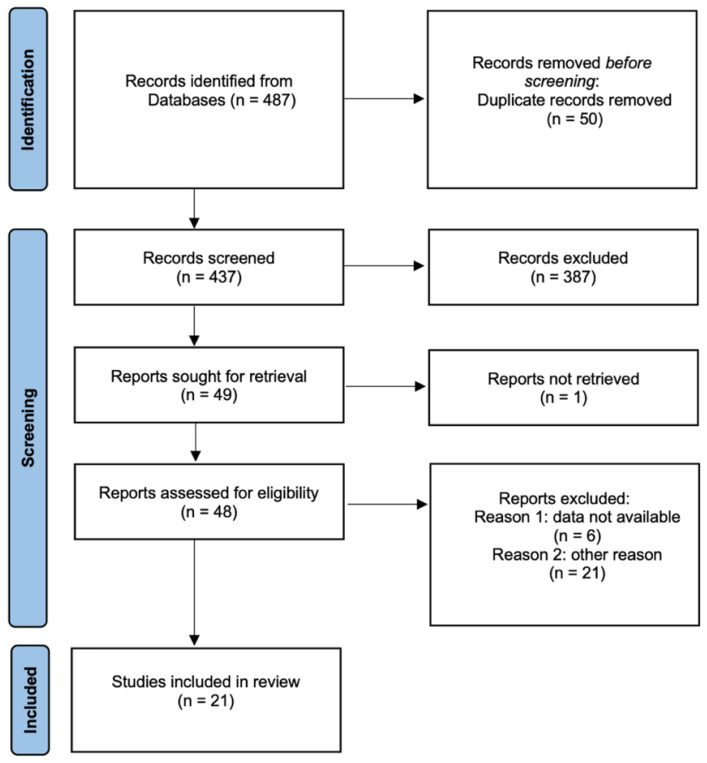
Preferred Reporting Items for Systematic Reviews and Meta-Analyses (PRISMA) flow diagram showing the selection process for including studies [41].

**Figure 2 cancers-15-03339-f002:**
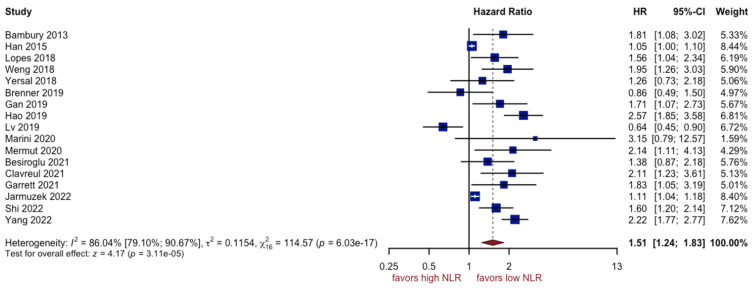
Forest plot of NLR [7,12,14,19,45,46,47,48,49,50,51,53,54,56,57,58,61].

**Figure 3 cancers-15-03339-f003:**
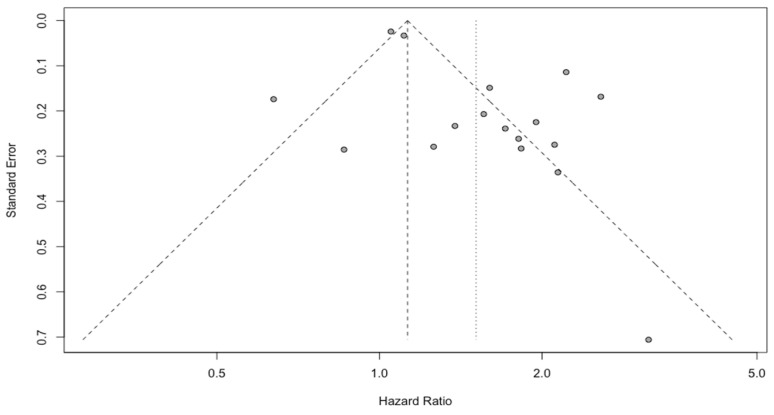
Funnel plot of NLR.

**Figure 4 cancers-15-03339-f004:**
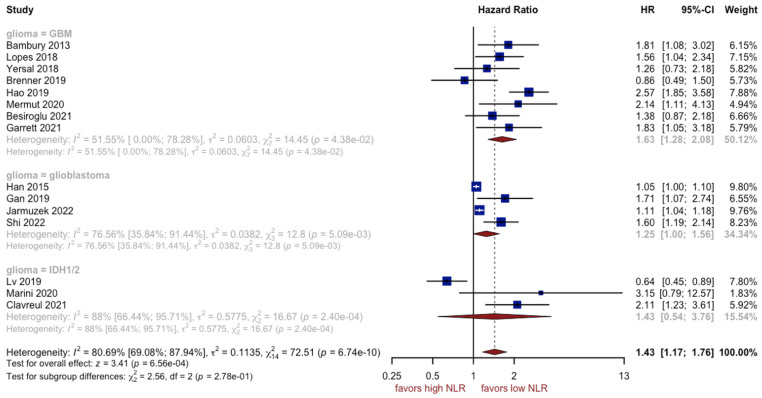
Forest plot of NLR for subgroups [7,12,14,45,46,47,48,49,50,51,53,54,56,57,58].

**Figure 5 cancers-15-03339-f005:**
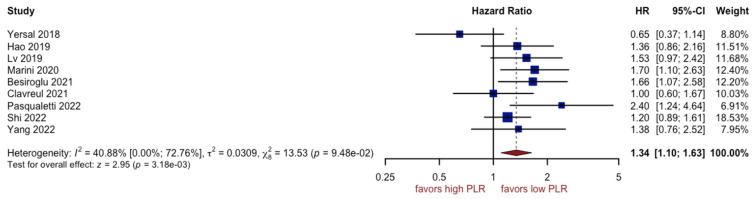
Forest plot of PLR [12,14,48,50,53,56,57,59,61].

**Figure 6 cancers-15-03339-f006:**
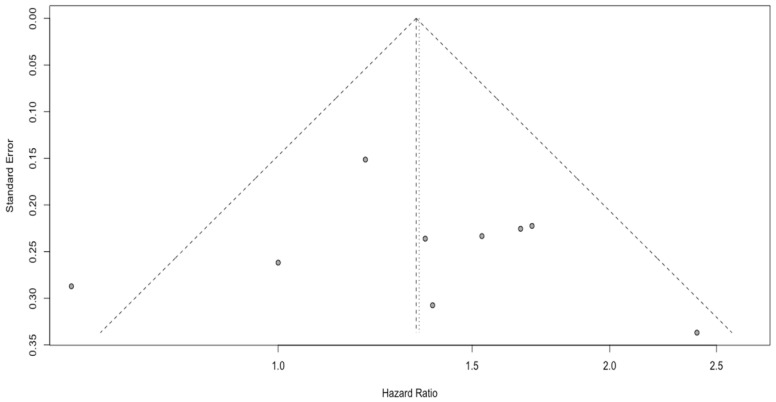
Funnel plot of PLR.

**Figure 7 cancers-15-03339-f007:**
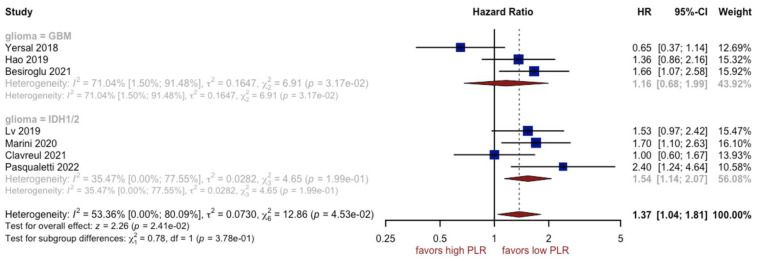
Forest plot of PLR for subgroups [14,48,50,53,56,57,59].

**Figure 8 cancers-15-03339-f008:**
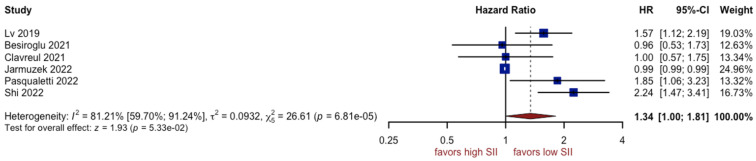
Forest plot of SII [7,12,14,56,57,59].

**Figure 9 cancers-15-03339-f009:**
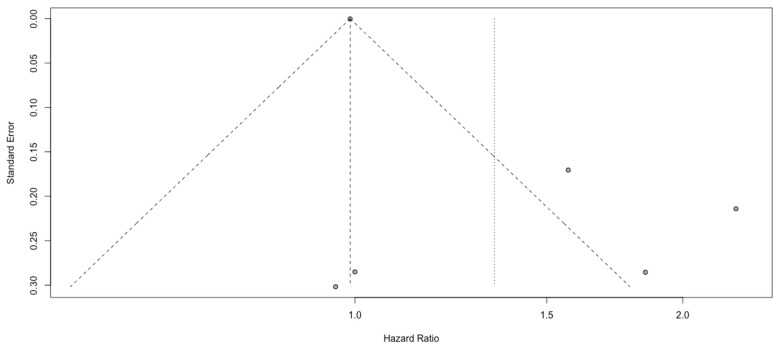
Funnel plot of SII.

**Figure 10 cancers-15-03339-f010:**
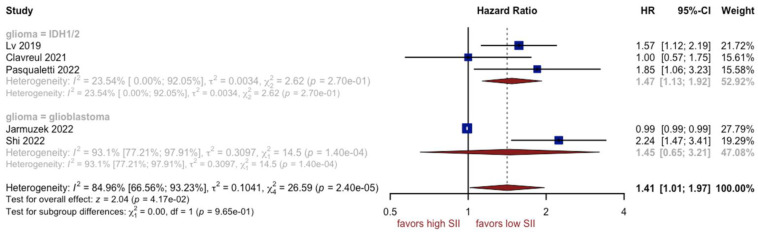
Forest plot of SII for subgroups [7,12,14,57,59].

**Figure 11 cancers-15-03339-f011:**
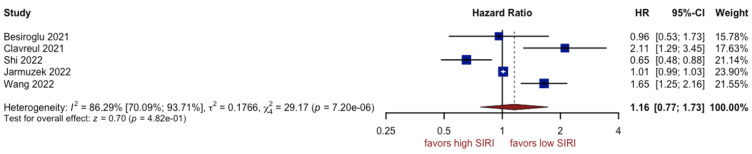
Forest plot of SIRI [7,12,56,57,60].

**Figure 12 cancers-15-03339-f012:**
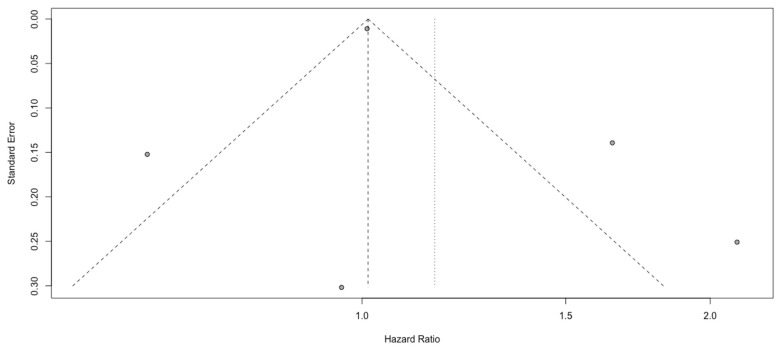
Funnel plot of SIRI.

**Figure 13 cancers-15-03339-f013:**
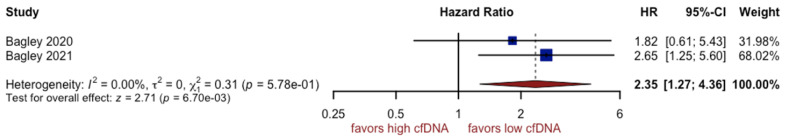
Forest plot of cfDNA [52,55].

**Table 1 cancers-15-03339-t001:** Search strategy for PubMed/Medline; MeSH and free text terms are marked with the syntax (Mesh) and (tiab), respectively.

Search #	Search Strategy	Items Found
1	“Glioblastoma” (Mesh)	32,216
2	“Glioblastoma/blood” (Mesh)	326
3	“Glioblastoma/immunology” (Mesh)	1300
4	“Glioblastoma/mortality” (Mesh)	2900
5	#1 AND (#2 OR #3 OR #4)	4340
6	“NLR” (tiab)	13,458
7	“PLR” (tiab)	5803
8	“SII” (tiab)	3635
9	“SIRI” (tiab)	591
10	“Cell-free DNA” (tiab)	6465
11	#5 AND (#6 OR #7 OR #8 OR #9 OR #10)	29

**Table 2 cancers-15-03339-t002:** The list of publications included in this meta-analysis.

Study	Duration	Sample Size Female/Male	NLR Cut-Off Value [10^3^/µL]	PLR Cut-Off Value [10^3^/µL]	SII Cut-Off Value [10^3^/µL]	SIRI Cut-Off Value [10^3^/µL]	cfDNA Cut-Off Value [ng/mL]	Outcome	Glioblastoma Type
Bambury 2013 [45]	2004–2009	74 (19/65)	4	-	-			multivariate	GBM
Han 2015 [46]	2010–2014	152 (57/95)	4	-	-	-	-	multivariate	grade IV
Lopes 2018 [47]	2005–2013	140 (42/98)	5	-	-	-	-	multivariate	GBM
Weng 2018 [19]	2011–2014	239 (108/131)	4	-	-	-	-	multivariate	44% IV grade
Yersal 2018 [48]	2012–2017	80 (41/39)	4	135	-	-	-	univariate	GBM
Brenner 2019 [49]	2005–2016	89 (43/46)	4	-	-	-	-	multivariate	GBM
Hao 2019 [50]	2012–2017	187 (71/116)	4.1	228.6	-	-	-	uni/multivariate	GBM
Gan 2019 [51]	2014–2018	135 (48/89)	3	-	-	-	-	multivariate	84% IV grade
Lv 2019 [14]	2006–2018	192 (79/113)	2.7	87	718	-	-	uni/multivariate	IDH -1 mutant and wild-type
Bagley 2020 [52]	-	42 (20/22)	-	-	-	-	13.4	multivariate	IDH 1 and 2 wild-type glioblastomas (83%)
Marini 2020 [53]	2013–2019	124 (65/59)	4	175	-	-	-	multivariate	IDH -1 mutant (48%) and wild-type (52%)
Mermut 2020 [54]	2011–2018	75 (28/47)	4	-	-	-	-	univariate	GBM
Bagley 2021 [55]	2018–2020	62 (23/39)	-	-	-	-	25.2	multivariate	IDH wild-type glioblastomas
Besiroglu 2021 [56]	2014–2019	107 (49/58)	2.9	159	785	-	-	multivariate	GBM
Clavreul 2021 [57]	2012–2020	85 (20/65)	2.42	180.9	502.39	2.55	-	univariate	IDH-wildtype
Garrett 2021 [58]	2013–2019	87 (33/54)	5.07	-	-	-	-	univariate	GBM
Jarmuzek 2022 [7]	2004–2021	358 (195/163)	4.56	-	2003	3.03	-	multivariate	66% IV grade
Pasqualetti 2022 [59]	2010–2020	77 (34/43)	-	250	1200	-	-	univariate	IDH 1 and 2 wild-type glioblastomas
Shi 2022 [12]	2014–2018	132 (105/127)	2.54	158.56	659.1	1.78	-	uni/multivariate	grade IV
Wang 2022 [60]	2015–2019	291 (105/106)	4.86	-	-	1.26	-	multivariate	grade IV
Yang 2022 [61]	2016–2019	187 (76/111)	2	213	-	-	-	uni/multivariate	47% IV grade

Abbreviations: NLR neutrophil-to-lymphocyte ratio, PLR platelet-to-lymphocyte ratio, SII systemic immune inflammation index, SIRI systemic inflammation response index, cfDNA cell-free DNA, GBM glioblastoma, IDH1 isocitrate dehydrogenase 1.

**Table 3 cancers-15-03339-t003:** NOS study assessment.

Study	Marker	S1	S2	S3	S4	C1	O1	O2	O3	Total
Bambury 2013 [45]	NLR	*	*	*	0	*	*	*	*	7
Han 2015 [46]	NLR	*	*	*	0	**	*	*	*	8
Lopes 2018 [47]	NLR	*	*	*	0	*	*	*	*	7
Weng 2018 [19]	NLR	*	*	*	0	*	*	*	*	7
Yersal 2018 [48]	NLR, PLR	*	*	*	0	**	*	*	*	8
Brenner 2019 [49]	NLR	*	*	*	0	**	*	*	*	8
Hao 2019 [50]	NLR, PLR	*	*	*	0	**	*	*	*	8
Gan 2019 [51]	NLR	*	*	*	0	*	*	*	*	7
Lv 2019 [14]	NLR, PLR, SII	*	*	*	0	**	*	*	*	8
Bagley 2020 [52]	cfDNA	*	*	*	0	*	*	*	*	7
Marini 2020 [53]	NLR, PLR	*	*	*	0	**	*	*	*	8
Mermut 2020 [54]	NLR	*	*	*	0	**	*	*	*	8
Bagley 2021 [55]	cfDNA	*	*	*	0	*	*	*	*	7
Besiroglu 2021 [56]	NLR, PLR, SII	*	*	*	0	*	*	*	*	7
Clavreul 2021 [57]	NLR, PLR, SII, SIRI	*	*	*	0	**	*	*	*	8
Garrett 2021 [58]	NLR	*	*	*	0	*	*	*	*	7
Jarmuzek 2022 [7]	NLR, SII, SIRI	*	*	*	0	**	*	*	*	8
Pasqualetti 2022 [59]	PLR, SII	*	*	*	0	**	*	*	*	8
Shi 2022 [12]	NLR, PLR, SII, SIRI	*	*	*	0	**	*	*	*	8
Wang 2022 [60]	NLR, SIRI	*	*	*	0	**	*	*	*	8
Yang 2022 [61]	NLR, PLR	*	*	*	0	**	*	*	*	8

Abbreviations: Selection (max *), S1 representativeness of the exposed cohort, S2 selection of the non-exposed cohort, S3 ascertainment of exposure, S4 demonstration that outcome of interest was not present at start of study. Comparability (max **), C1 comparability of cohorts based on the design or analysis. Outcome (max *), O1 assessment of outcome, O2 follow-up long enough for outcomes to occur, O3 adequacy of follow-up of cohorts, NLR neutrophil-to-lymphocyte ratio, PLR platelet-to-lymphocyte ratio, SII systemic immune inflammation index, SIRI systemic inflammation response index, cfDNA cell-free DNA.

## Data Availability

The data presented in this study are available on request from the corresponding author.

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
