# Peer review of "Prognostic Values of Systemic Inflammatory Immunological Markers in Glioblastoma: A Systematic Review and Meta-Analysis"

_cancers, 2023, doi:10.3390/cancers15133339_

Round 1
Reviewer 1 Report
The manuscript was prepared very well. The introduction section justifies the purpose of the study. I congratulate the authors for the preparation of the manuscript
I would like to congratulate the authors for the structure of the manuscript and all the research carried out. It is highly publishable. However, there are some concerns, in part
Introduction
· Why is this study considered relevant?
· I suggest that incorporate more background information on the objective of the study
Methods
· please include an assessment of the methodological quality by the PEDRo or McMaster standards.
· It would be appropriate to carry out an analysis of biases by Crochane rules.
Results
· It is the strong part of the manuscript does not require any changes
· Incorporate the abbreviations used in the footers of the tables
Discussion
· Include a section on strengths / limitations.
· What does this article contribute to, the authors should make their own assessment and include their own discussion of the results shown in the manuscript?
· include a section on future scenarios
Conclusion
· In the Conclusion section, state the most important outcome of your work. Do not simply summarize the points already made in the body — instead, interpret your findings at a higher level of abstraction. Show whether, or to what extent, you have succeeded in addressing the need stated in the Introduction (or objectives).
Reviewer 2 Report
A very interesting meta-analysis that summarizes the findings in the field of glioblastoma. Extensive discussion of the results, taking into account the limitations of the analyzes performed.
Reviewer 3 Report
After a careful reading of the manuscript entitled” Prognostic values of systemic inflammatory immunological markers in glioblastoma a systematic review and meta-analysis”, I have a few changes that need to be incorporated in the manuscript. Overall the manuscript has good literature analysis
1. There are some parameters used such as neutrophil-to-lymphocyte ratio (NLR), platelet-to- 14 lymphocyte ratio (PLR), systemic immune inflammation index (SII) and systemic inflammation 15 response index (SIRI) as prognostic factors in patients with glioblastoma. Why other factors such as PDL1, Isocitrate dehydrogenase 1 (IDH 1) and O6-methylguanine-DNA methyltransferase (MGMT) was not considered in the studies?
2. What were the The Mesh Terms and free terms included in the meta-analysis. A detailed information should be provided in the supplementary files.
3. The authors should include quality evaluations and exclusion and inclusions associated with the studies in material and method section
4. The manuscript has some grammatical errors that can be improved.
5. Result section should be explain in elaborative form.
The manuscript has some grammatical errors that can be improved.
